# EFFICIENT BAYESIAN DNN COMPRESSION THROUGH SPARSE QUANTIZED SUB-DISTRIBUTIONS

## ABSTRACT

This paper presents a novel method that simultaneously achieves model pruning and low-bit quantization through Bayesian variational inference to effectively compress deep neural networks (DNNs) while suffering minimal performance degradation. Unlike previous approaches that treat pruning and quantization as separate, sequential tasks, our method explores a unified optimization space, enabling more efficient compression. By leveraging a spike-and-slab prior combined with Gaussian Mixture Models (GMM), we can achieve both network sparsity and low-bit representation. Experiments on CIFAR-10, CIFAR-100, and SQuAD datasets demonstrate that our approach achieves compression rates of up to 32x with less than a $1.3\%$ accuracy loss on the CIFAR datasets and a 1.66 point decrease in F1 score on SQuAD. Additionally, we show that the Bayesian model average of neural networks can further mitigate the impact of quantization noise, leading to more robust compressed models. Our method outperforms existing techniques in both compression efficiency and accuracy retention, offering a promising solution for compressing DNNs.

## 1 INTRODUCTION

Deep Neural Networks (DNNs) have emerged as a leading approach in various machine learning tasks due to their superior performance across domains such as computer vision (He et al., 2016; 2017; Dosovitskiy et al., 2021), natural language processing (Devlin, 2018; Xu et al., 2020; Touvron et al., 2023), and speech recognition (Hinton et al., 2012; Zhang et al., 2023). However, this remarkable performance comes with a significant increase in computational and memory demands (Simonyan & Zisserman, 2014; He et al., 2016; Vaswani, 2017; Radford, 2018; Xu et al., 2020; Touvron et al., 2023). Various techniques like pruning (LeCun et al., 1989; Han et al., 2015), weight quantization (Courbariaux et al., 2015; Rastegari et al., 2016; Sze et al., 2017; Frantar et al., 2022; Lin et al., 2024), knowledge distillation (Park et al., 2019; Gou et al., 2021) and neural architecture search (Liu et al., 2018a;b; Wang et al., 2020b) have been proposed to improve DNNs efficiency and enhance the widespread of DNNs in AI systems.

Model compression techniques, including pruning and quantization, have proven effective in deploying cost-efficient DNNs (Buciluă et al., 2006; Choudhary et al., 2020). Pruning involves selectively removing DNN connections (i.e., setting the corresponding weights to zero), whereas weight quantization entails reducing the bit-width of weight representations. Pruning methods are typically categorized into structural pruning (Ding et al., 2019; You et al., 2019), which zeroes out groups of weights, and unstructured pruning (Guo et al., 2016; Dong et al., 2017), which zeroes out individual weights without altering the model's architecture. As for the quantization method, recent studies (Wang et al., 2018; Banner et al., 2018; Sun et al., 2019) have demonstrated that under 8-bit training techniques, it can effectively accelerate the training of various models, including VGG (Wu et al., 2018), ResNet (Banner et al., 2018), LSTMs, Transformers (Sun et al., 2019), and vision-language models (Wortsman et al., 2023).

Han et al. (2015) proposed a model compression pipeline that sequentially applies pruning and weight quantization, achieving significant compression rates without sacrificing much accuracy, however, the sequential application fails to explore the complementarity of pruning and quantization Bai et al. (2023). Recent studies have demonstrated that integrating pruning and quantization into a single process not only conserves computational resources but also achieves state-of-the-art

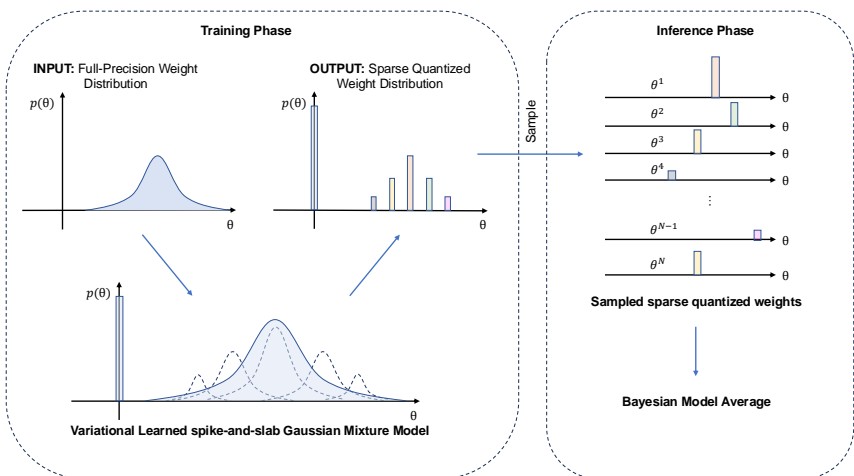

Figure 1: Variational learning of a sparse quantized weight sub-distributions as described in equation (8), from which sparse and quantized weights are sampled. Then sampled weights are ensembled via Bayesian model averaging to improve the model robustness to quantization noise.

performance (Van Baalen et al., 2020; Wang et al., 2020b; Frantar & Alistarh, 2022; Bai et al., 2022). Following this line of research, we propose a novel joint pruning and quantization method that statistically explores compressed DNNs via variational inference.

In this paper, we introduce the **S**parse **Q**uantized **S**ub-distribution (**SQS**) compression method, a novel approach that unifies pruning and quantization by identifying the optimal sparse quantized sub-distribution and enhancing resilience to performance degradation through Bayesian model averaging. Compared to previous efforts (e.g., Frantar & Alistarh, 2022; Gil et al., 2021), our approach introduces a novel Bayesian method that unifies the search spaces of both pruning and quantization. Existing approaches (Frantar & Alistarh, 2022; Gil et al., 2021) design separate solvers, that pursue (greedy) pruning and (greedy) quantization respectively, and combine these two by alternately applying the pruning and quantization solvers. Recognizing the untapped potential in optimizing the quantization procedure simultaneously with the pruning procedure, our method as shown in Figure 1 integrates the pruning and quantization process to identify the optimal sparse quantized sub-distribution that best approximates the original dense, full-precision weight distribution of DNNs. Moreover, as shown by previous works (Zhang et al., 2022b; Wang et al., 2024), Bayesian deep neural networks can offset the performance degradation resulting from the DNNs weight precision loss introduced by the quantization function, leading to more robust performance. Therefore, we leverage the power of variational learning to solve the sub-distribution approximation problem and facilitate Bayesian deep neural network training, and our solution achieves a significant compression rate with minimal impact on performance. Our code is available at `https://anonymous.4open.science/r/SQS-68EE/`.

## 2 RELATED WORKS

### 2.1 PRUNING AND SPARSE DNN

The concept of weight pruning was initially introduced by LeCun et al. (1989), with further development by Hassibi et al. (1993) through a mathematical method known as the Optimal Brain Surgeon (OBS). This approach selects weights for removal from a trained neural network using second-order information. Subsequent improvements, as indicated by studies (Dong et al., 2017; Wang et al., 2019; Singh & Alistarh, 2020), have adapted OBS for large-scale DNNs by employing numerical techniques to estimate the second-order information required by OBS for extensive model parameters. Meanwhile, Louizos et al. (2018b) has introduced an $l_0$ regularized method to enhance sparsity in DNNs. Frankle & Carbin (2019) established a critical insight that within a randomly initialized DNN, an optimal sub-network can be identified and extracted. More recently, amidst the rise of large

language models (LLMs), the work by Xia et al. (2024) has illustrated that structured pruning, combined with targeted retraining, can significantly reduce computational costs while preserving robust performance. Concurrently, researchers (Deng et al., 2019; Blundell et al., 2015; Bai et al., 2020) have employed spike-and-slab distributions and show the power of Bayesian Neural Networks in promoting sparsity in DNNs. Their efforts include comprehensive theoretical analysis that bridges the theoretical foundations with practical applications, thus advancing our understanding of model efficiency. Empirically, many of the aforementioned methods would require incremental pruning followed by retraining to preserve satisfactory performance.

## 2.2 Quantization

Quantization has emerged as a pivotal technique for enhancing the efficiency of deep neural networks (DNNs) (Sze et al., 2017; Frantar et al., 2022; Lin et al., 2024). Research in this domain generally follows two approaches: discontinuous-mapping quantization (Gupta et al., 2015; Hubara et al., 2018; Wu et al., 2018) and continuous-mapping quantization (Louizos et al., 2017; Ullrich et al., 2017; Dong et al., 2022; Shayer et al., 2018; Roth & Pernkopf, 2018). Discontinuous quantization involves a rounding function that projects full-precision weights onto a low-bit grid (Gupta et al., 2015; Wu et al., 2018; Louizos et al., 2018a; Hubara et al., 2018; Courbariaux et al., 2015; De Sa et al., 2018; Marchesi et al., 1993). To address the non-differentiability of discontinuous-mapping quantization, researchers have adopted the *straight through estimator* (STE) to facilitate backpropagation in networks with quantized, discrete weights (Courbariaux & Bengio, 2016; Courbariaux et al., 2015; Hubara et al., 2018; Rastegari et al., 2016). However, the STE can generate pseudo-gradients that may deviate weights from optimal values and increase training instability (Yin et al., 2019). Meanwhile, many researchers propose post-training quantization methods that have limited access to the training dataset (Wang et al., 2020a; Hubara et al., 2021; Li et al., 2021; Frantar & Alistarh, 2022; Frantar et al., 2022; Lin et al., 2024). BitSplit (Wang et al., 2020a) incrementally constructs quantized values using a squared error metric based on residual errors. In contrast, AdaQuant (Hubara et al., 2021) utilizes STE for direct optimization. BRECQ (Li et al., 2021) integrates Fisher information into the optimization process and focuses on the joint optimization of layers within individual residual blocks. Extending the Optimal Brain Surgeon (OBS) framework, Exact Optimal Brain Quantization (OBQ) (Frantar & Alistarh, 2022) adapts second-order weight pruning methods to quantization tasks. With the rise of LLMs demanding substantial computational resources, GPTQ (Frantar et al., 2022) employs second-order information for error compensation on calibration sets to speed up generative models. Additionally, AWQ (Lin et al., 2024) implements activation-aware quantization, selectively bypassing the quantization of key weights.

In contrast to the discontinuous-mapping quantization, continuous-mapping quantization avoids pseudo-gradients and thus would provide a more stable and accurate solution (Yin et al., 2019). Various studies have established specific prior distributions to approximate the quantized discrete distribution through variational learning (Ullrich et al., 2017; Louizos et al., 2017; Shayer et al., 2018) and Markov Chain Monte Carlo (MCMC) methods (Roth & Pernkopf, 2018). However, these methods either need manual setting of priors (Ullrich et al., 2017; Louizos et al., 2017; Shayer et al., 2018) or would increase memory footprint (Roth & Pernkopf, 2018). DGMS (Dong et al., 2022) is an automated quantization method that utilizes Gaussian Mixture that avoids the aforementioned problem. Our method is similar to the DGMS (Dong et al., 2022), but further, enhances the compression rate by unifying pruning and quantization, and boosts performance by utilizing the property that the Bayesian average of DNNs are particularly robust to the quantization noise (Wang et al., 2024).

## 3 Preliminary

### 3.1 Quantization

A quantization function can be presented as $Q : x \in \mathbb{R} \to \mathcal{Q} = \{\mu_1, \ldots \mu_K\}$, where $x$ is the real-valued number and $\mathcal{Q}$ denotes the set of discrete representation after quantization. For example, given a stepsize $\Delta$, a symmetric quantization function $Q_d$ maps a full-precision number to its nearest low-bit representable neighbor within the range $[-K\Delta, K\Delta]$ as follows:

$$Q_d(x) = \text{sign}(x) \cdot \min\left( \Delta \left\lfloor \frac{|x|}{\Delta} + \frac{1}{2} \right\rfloor, K\Delta \right).$$

Meanwhile, a naive stochastic quantization function has the following form:

$$Q_s(x) = \begin{cases} \Delta \left\lfloor \frac{x}{\Delta} \right\rfloor, & \text{w.p. } \left\lceil \frac{x}{\Delta} \right\rceil - \frac{x}{\Delta} \\ \Delta \left\lceil \frac{x}{\Delta} \right\rceil, & \text{w.p. } 1 - \left( \left\lceil \frac{x}{\Delta} \right\rceil - \frac{x}{\Delta} \right). \end{cases}$$

This stochastic quantization preserves more information because $\mathbb{E}[Q_s(x)] = x$, a property that is particularly advantageous when $x$ is close to zero, as it prevents the value from being consistently quantized to zero, unlike deterministic quantization $Q_d$. Given the observations of the clustering effect of DNNs weights (Han et al., 2015), DGMS (Dong et al., 2022) has proposed a trainable quantization method, where each weight is quantized to one of the representations in the adaptive quantization set $\mathcal{Q}_A = \{\mu_1, \cdots, \mu_K\}$ where $\mu_k \in \mathbb{R}$ is also trained within the overall optimization process. Let $\theta$ denote the weights vector in $\mathbb{R}^T$ indicating the set of weights, with $T$ being the total number of weights in the DNN. Rather than storing $T$ full-precision weights, the weights are quantized into a few discrete values (i.e., shared full precision value), and only a small index indicating which shared value in $\mathcal{Q}_A$ is assigned is stored for each weight, where the $T$ is the total number of weights. This technique not only reduces memory footprint but also accelerates DNN inference through caching and weight reuse (Dong et al., 2022; Han et al., 2015; Xiao et al., 2019). Trivially, a smaller $K$ results in higher quantization noise; on the other hand, when $K$ is as large as the total number of DNN weights, the quantization set $\mathcal{Q}_A$ can accurately replicate the full-precision DNN weights under appropriate settings. Note that inevitably, any quantization function introduces noise to the DNN weights, i.e., the gap between the full-precision number and its quantized value, hence harming the predictive performance. In addition, the discontinuity of the quantization mapping suffers from non-differentiability, posing difficulties to the optimization process.

## 3.2 Variational Learning

Given the observed dataset $\mathcal{D}$, a Bayesian procedure aims to infer from the true posterior distribution $\pi(\theta|\mathcal{D}) \propto \pi(\theta)p(\mathcal{D};\theta)$, where $\pi(\theta)$ is the prior and $p(\mathcal{D};\theta)$ is the likelihood. Since the posterior is usually intractable, variational inference (Jordan et al., 1999; Blei et al., 2017) tries to approximate the true distribution by the closest member in terms of Kullback–Leibler (KL) divergence (Csiszár, 1975) from the variational family of distributions $\mathcal{F}$:

$$q^*(\theta) = \underset{q(\theta) \in \mathcal{F}}{\arg\min} \, D_{\mathrm{KL}}\left(q(\theta) || \pi(\theta|\mathcal{D})\right). \tag{1}$$

The optimization (1) is equivalent to minimize the negative *Evidence Lower Bound* (ELBO) defined as:

$$\Omega = -\mathbb{E}_{q(\theta)}[\log p(\mathcal{D};\theta)] + D_{\mathrm{KL}}(q(\theta) || \pi(\theta)), \tag{2}$$

where the first term in (2) is the expected log-likelihood which measures how well the variational distribution $q(\theta)$ aligns with the likelihood of the observed data. The expected log-likelihood usually cannot be integrated analytically and thus we employ a further soft-max approximation described in the later context. The second term works as regularization, and by setting a spike-and-slab prior distribution, it can promote and enforce sparsity in the weight distribution, encouraging the model to favor sparse solutions.

## 4 Methodology

In this section, we first reformulate the traditional weight quantization function and then we propose a novel spike-and-slab-like variational family to model sparse quantized distributed DNNs. Finally, we present a Bayesian algorithm to unify the pruning and task-optimal weight quantization process.

## 4.1 Quantized Sub-distribution

Let $f(\cdot, \theta)$ represent the deep neural network. Here, $\theta_i$ denotes the $i$-th component of the weight vector $\theta$. Given a quantization set $\mathcal{Q}$, we define an adaptive stochastic quantization mapping $Q : \theta \in \mathbb{R} \to \mathcal{Q} = \{\mu_1, \ldots, \mu_K\}$ as:

$$Q(\theta_i) = \mu_k, \quad \text{w.p. } p_{ki}, \text{ for } k = 1, \ldots, K \text{ and } i = 1, \ldots, T. \tag{3}$$

However, learning quantization functions for all $T$ weights is computationally infeasible (given that a typical DNN model contains millions or billions of weights), and direct gradient-based optimization for discrete quantization functions is also challenging. To address this, we approximate the multinomial distribution of the quantized weight $Q(\theta_i)$ with a Gaussian Mixture Model (GMM)

$$g\left((\mu_1, \sigma_1^2), \cdots, (\mu_K, \sigma_K^2), \theta_i\right) \sim \sum_{k=1}^{K} \phi_k(\theta_i)\mathcal{N}(\mu_k, \sigma_k^2). \tag{4}$$

where $\mathcal{N}(\mu, \sigma^2)$ denotes the Gaussian random variable, $\theta_i$ is the full-precision preimage weight, and the mixture weight $\phi_k(\theta)$ has a parametric form. Note that with slight abuse of notation, we also use $\mathcal{N}(\cdot|\mu, \sigma^2)$ to represent the Gaussian density function. Inspired by the connection between the clustering problem and the Gaussian mixture modeling, we let $\phi_k(\theta_i)$ be related to the posterior probability of the weight $\theta_i$ sampled from the Gaussian components $\mathcal{N}(\mu_k, \sigma_k^2)$. That is, given a prior distribution $\varpi = [\varpi_1, \ldots, \varpi_K]$ over the quantization set $\mathcal{Q}$, the posterior component weight $\varphi_k(\theta_i)$ is:

$$\varphi_k(\theta_i) = \varphi_k(\theta_i; \varpi) = \frac{\exp\left(\varpi_k \mathcal{N}(\theta_i|\mu_k, \sigma_k^2)\right)}{\sum_{j=1}^{K} \exp\left(\varpi_j \mathcal{N}(\theta_i|\mu_j, \sigma_j^2)\right)}. \tag{5}$$

Given a temperature parameter $\tau_1$, we further define $\phi_k$ via temperature-based softmax as

$$\phi_k(\theta_i) = \phi_k(\theta_i; \varpi, \tau_1) = \frac{\exp\left(\varphi_k(\theta_i)/\tau_1\right)}{\sum_{j=1}^{K} \exp\left(\varphi_j(\theta_i)/\tau_1\right)}, \text{ for } k = 1, \ldots, K, \tag{6}$$

such that $\phi_k$'s and $\psi_k$'s share the same numerical order, and parameter $\tau_1$ grants trainable controls on the distribution concentration, i.e., as $\tau_1 \to 0$, (4) reduces to a single normal distribution. Notice that with small enough $\sigma_i^2$, the Gaussian Mixture Model in (4) reduces to a multinomial distribution over $\mathcal{Q}$. It is worth mentioning that DGMS (Dong et al., 2022) utilizes the same mixture normal structure. But the GMM model merely serves as a clustering tool for DGMS, while our method adopts a more principled Bayesian modeling approach, laying the foundation for Bayesian model averaging which could improve robustness to quantization noise. Furthermore, by building on the GMM approximation, a sparse distribution can be seamlessly integrated, forming a unified search space for both quantization and pruning, which may lead to a globally optimal solution.

## 4.2 SQS: Sparse Quantized Sub-distribution

In this section, we introduce a novel unified pruning and quantization method by finding a sparse quantized sub-distribution via variational learning. The ultimate goal is to approximate dense full-precision DNNs denoted as $f(\cdot; \theta)$, with Bayesian sparse and low-precision counterparts $f(\cdot, \widetilde{\theta})$. To achieve this goal, we utilize a spike-and-slab prior (Ishwaran & Rao, 2005; Bai et al., 2020) incorporated with a Gaussian Mixture distribution to represent a sparse, quantized weight sub-distribution.

A Dirac distribution located at zero and a flat slab distribution constitute the spike-and-slab which is utilized to enforce sparsity in DNNs (Bai et al., 2020). With $\delta_0$ denoting the Dirac distribution centered at zero, and $\gamma = (\gamma_1, \cdots, \gamma_T)$ with each $\gamma_i$ binary random variable representing whether the weight $\theta_i$ is selected to be pruned or not, the spike-and-slab prior is defined as:

$$\tilde{\theta}_i|\gamma_i \sim \gamma_i \mathcal{N}(0, \sigma_0^2) + (1 - \gamma_i)\delta_0, \quad \gamma_i \sim \text{Bern}(\lambda),$$

for $i = 1, \cdots, T$, where $\sigma_0^2$ and $\lambda$ are the hyperparameters representing prior sparsity level and prior Gaussian variance. By simply integrating out the variable $\gamma_i$, one can derive the marginal prior distribution $\pi(\tilde{\theta}_i)$ as:

$$\lambda\mathcal{N}(0, \sigma_0^2) + (1 - \lambda)\delta_0. \tag{7}$$

The parameter $1 - \lambda$ represents the prior probability that a weight will be pruned. For instance, in a DNN with a sparsity level of 90%, $\lambda$ would be set to 0.1, resulting in $1 - \lambda = 0.9$, indicating a 90% prior chance that a given weight will be pruned. We then design a novel spike-and-slab with a Gaussian Mixture Model variational family to model the sparse quantized posterior weight distribution. Given the GMM in (4), one natural idea is to combine this distribution with Dirac distribution $\delta_0$ to form a variational family $\mathcal{F}$. That is, any $q(\theta) \in \mathcal{F}$ has the following form:

$$\tilde{\theta}_i|\gamma_i \sim \gamma_i g\left((\mu_1, \sigma_1^2), \cdots, (\mu_K, \sigma_K^2), \theta_i\right) + (1 - \gamma_i)\delta_0,$$

$$\gamma_i \sim \text{Bern}(\tilde{\lambda}_i), \text{ for } i = 1, \ldots, T.$$

To make the sub-distribution fully learnable with gradient, we reparameterize $\tilde{\lambda}_i$ as follows:

$$\tilde{\lambda}_i = \frac{\exp\left(\tilde{s}_i/\tau_2\right)}{1 + \exp\left(\tilde{s}_i/\tau_2\right)},$$

for $i = 1, \ldots, T$, where $\tilde{s}_i$ is an auxiliary variable and $\tau_2$ is a temperature to facilitate the training process. Similar to (7), we can get the marginal variational distribution $q(\tilde{\theta}_i)$ as:

$$\tilde{\lambda}_i g\left((\mu_1, \sigma_1^2), \cdots, (\mu_K, \sigma_K^2), \theta_i\right) + (1 - \tilde{\lambda}_i)\delta_0. \tag{8}$$

Finally, the variation learning aims to minimize the ELBO defined as the following:

$$\Omega = -\mathbb{E}_{q(\tilde{\theta})}\left[\log p(\mathcal{D}; \tilde{\theta})\right] + D_{\mathrm{KL}}\left(q(\tilde{\theta})||\pi(\tilde{\theta})\right)$$

$$= -\mathbb{E}_{q(\tilde{\theta})}\left[\log p(\mathcal{D}; \tilde{\theta})\right] + \sum_{i=1}^{T} D_{\mathrm{KL}}\left(q(\tilde{\theta}_i)||\pi(\tilde{\theta}_i)\right). \tag{9}$$

---

**Algorithm 1** Variational Learning Sparse & Quantized Sub-distribution

---

**Input:** Training dataset $\mathcal{D} = \{\mathcal{X}, \mathcal{Y}\}$, DNN $f(\cdot; \theta)$ of with full-precision initial weights $\theta \in R^T$, GMM component number $K$, initial temperature $\tau_1, \tau_2$ and prior variance $\sigma_0^2$.

1: **Initialization**
2: $\mathcal{R} \leftarrow \{\theta | \theta \in \text{region } k\}_{k=0}^{K}$; ▷ initial region generation with k-means

3: $\vartheta \leftarrow \left\{\hat{\mu}_k, \hat{\varpi}_k \leftarrow \frac{|\mathcal{R}_k|}{|\theta|}, \hat{\sigma}_k \leftarrow \sqrt{\frac{\sum_{j=1}^{T}(\theta_j - \hat{\mu}_k)^2}{|\theta|-1}}\right\}_{k=0}^{K}$;

4: **Training**
5: **while** not converged **do**
6:    **for** $k \leftarrow 1$ **to** $K$ **do**
7:       $\phi_k(\theta; \hat{\varpi}, \tau_1) \leftarrow \frac{\exp\left(\varphi_k(\theta)/\tau_1\right)}{\sum_{i=1}^{K}\exp\left(\varphi_i(\theta/\tau_1)\right)}$, Eqn. (4) and Eqn. (6);
8:    **end for**
9:    $\widetilde{\Phi}(\tilde{\theta}; \hat{\varpi}, \tau_1) \leftarrow \tilde{\lambda}\sum_{k=1}^{K}\mu_k\phi_k(\theta; \hat{\varpi}, \tau_1)$, Eqn. (11);
10:   Calculate the relaxed ELBO $\widetilde{\Omega}$, Eqn (12);
11:   Backpropagation and update $\{\theta, \vartheta, \tilde{\lambda}\}$ with the stochastic gradient descent;
12: **end while**
**Output:** The sparse quantized weight sub-distribution $\hat{q}(\tilde{\theta})$.

---

**Approximation** It is important to note that the KL divergence between the variational distribution and the spike-and-slab prior distribution does not have a closed-form solution. To simplify the ELBO and validate our approach, we reformulate a key lemma from previous work (Chérief-Abdellatif & Alquier, 2018), as follows:

**Lemma 1.** *For any $K > 0$, the KL divergence between any two mixture densities $\sum_{k=1}^{K} w_k g_k$ and $\sum_{k=1}^{K} \tilde{w}_k \tilde{g}_k$ is bounded as*

$$D_{\mathrm{KL}}(\sum_{k=1}^{K} w_k g_k || \sum_{k=1}^{K} \tilde{w}_k \tilde{g}_k) \leq D_{\mathrm{KL}}(\boldsymbol{w}||\tilde{\boldsymbol{w}}) + \sum_{k=1}^{K} w_k D_{\mathrm{KL}}(g_k||\tilde{g}_k),$$

*where $D_{\mathrm{KL}}(\boldsymbol{w}||\tilde{\boldsymbol{w}}) = \sum_{k=1}^{K} w_k \log \frac{w_k}{\tilde{w}_k}$.*

Given the definitions in equation (7), (8) and Lemma 1, the ELBO can be further bounded as:

$$\Omega \leq -\mathbb{E}_{q(\tilde{\theta})}[\log p(\mathcal{D}; \tilde{\theta})] + \sum_{i=1}^{T}\left(\tilde{\lambda}_i \log \frac{\tilde{\lambda}_i}{\lambda} + (1 - \tilde{\lambda}_i)\log\frac{1 - \tilde{\lambda}_i}{1 - \lambda}\right)$$

$$+ \sum_{i=1}^{T} \tilde{\lambda}_i D_{\mathrm{KL}}\left(g((\mu_1, \sigma_1^2), \cdots, (\mu_K, \sigma_K^2), \theta_i)||\mathcal{N}(0, \sigma_0^2)\right). \tag{10}$$

Again the KL divergence between the Gaussian Mixture Model and Gaussian distribution $D_{\text{KL}}(g((\mu_1, \sigma_1^2), \cdots, (\mu_K, \sigma_K^2), \theta_i) || \mathcal{N}(0, \sigma_0^2))$ does not have a closed form, but can be further upper bounded as:

$$D_{\text{KL}}(g((\mu_1, \sigma_1^2), \cdots, (\mu_K, \sigma_K^2), \theta_i) || \mathcal{N}(0, \sigma_0^2))$$

$$= D_{\text{KL}} \left( \sum_{k=1}^K \phi_k(\theta_i) \mathcal{N}(\mu_k, \sigma_k^2) || \sum_{k=1}^K \phi_k(\theta_i) \mathcal{N}(0, \sigma_0^2) \right)$$

$$\leq \sum_{k=1}^K \phi_k(\theta_i) D_{\text{KL}} \left( \mathcal{N}(\mu_k, \sigma_k^2) || \mathcal{N}(0, \sigma_0^2) \right),$$

where the last inequality is by Lemma 1. Combined with equation (10), the ELBO $\Omega$ can be bounded as:

$$\Omega \leq - \mathbb{E}_{q(\tilde{\theta})}[\log p(\mathcal{D}; \tilde{\theta})] + \sum_{i=1}^T \left( \tilde{\lambda}_i \log \frac{\tilde{\lambda}_i}{\lambda} + (1 - \tilde{\lambda}_i) \log \frac{1 - \tilde{\lambda}_i}{1 - \lambda} \right)$$

$$+ \sum_{i=1}^T \sum_{k=1}^K \phi_k(\theta_i) \tilde{\lambda}_i D_{\text{KL}}(\mathcal{N}(\mu_k, \sigma_k^2) || \mathcal{N}(0, \sigma_0^2)).$$

Beyond that, the first term $\mathbb{E}_{q(\tilde{\theta})}[\log p(\mathcal{D}; \tilde{\theta})]$ is also intractable. A common approach to approximate this term is Monte Carlo sampling James (1980), where samples are drawn directly from the distribution $q(\tilde{\theta})$ via the so-called reparameterization trick. However, this method requires massive computations to provide an accurate estimation. Instead, we consider the distribution mean of $q(\tilde{\theta})$

$$\widetilde{\Phi}(\tilde{\theta}_i; \varpi, \tau_1) = \tilde{\lambda}_i \sum_{k=1}^K \mu_k \phi_k(\theta_i; \varpi, \tau_1) + (1 - \tilde{\lambda}_i) * 0 = \tilde{\lambda}_i \sum_{k=1}^K \mu_k \phi_k(\theta_i; \varpi, \tau_1), \qquad (11)$$

and approximate first term of (9) $\mathbb{E}_{q(\tilde{\theta})}[\log p(\mathcal{D}; \tilde{\theta})]$ by $\log p(\mathcal{D}; \widetilde{\Phi}(\tilde{\theta}))$. That is, we approximate $q(\tilde{\theta})$ by a Delta measure on $\widetilde{\Phi}(\tilde{\theta}_i; \varpi, \tau_1)$. This approximation seems brutal, but works well in practice, as we notice that we need to pick a relatively small $\tau_1$ value to achieve a satisfactory performance, and $\sigma_k^2$'s usually converge to small values. Along with the small temperature $\tau_1$, $\phi_k(\theta_i), k = 1, \ldots, K$ converges to one-hot vector, thus $\sum_{k=1}^K \phi_k(\theta_i) D_{\text{KL}}(\mathcal{N}(\mu_k, \sigma_k^2) || \mathcal{N}(0, \sigma_0^2))$ is close to

$$\sum_{k=1}^K D_{\text{KL}}(\mathcal{N}(\mu_k, \sigma_k^2) || \mathcal{N}(0, \sigma_0^2)) * \mathcal{I}(k = \arg\max_k \phi_k(\theta_i)).$$

Finally, we define an approximate objective:

$$\widetilde{\Omega} = - \log p(\mathcal{D}; \widetilde{\Phi}(\tilde{\theta})) + \sum_{i=1}^T \left( \tilde{\lambda}_i \log \frac{\tilde{\lambda}_i}{\lambda} + (1 - \tilde{\lambda}_i) \log \frac{1 - \tilde{\lambda}_i}{1 - \lambda} \right)$$

$$+ \sum_{i=1}^T \sum_{k=1}^K D_{\text{KL}}(\mathcal{N}(\mu_k, \sigma_k^2) || \mathcal{N}(0, \sigma_0^2)) \mathcal{I}(k = \arg\max_k \phi_k(\theta_i)). \qquad (12)$$

We are now prepared to combine all components into a comprehensive training algorithm, as outlined in Algorithm 1.

**Inference** Let $\hat{q}(\cdot) \in \mathcal{F}$ denote the optimization solution of the above variational learning, associated with parameter estimations $\hat{\theta}_i, \hat{\mu}_i, \hat{\sigma}_i^2, \hat{\lambda}_i$ for $i = 1, \cdots, T$. In the inference stage, the sparse quantized weight can be sampled as the following:

$$\tilde{\theta}_i = \begin{cases} \hat{\mu}_k, & \text{w.p. } \phi_k(\hat{\theta}_i; \hat{\varpi}, \tau_1) \text{ for } k = 1, \ldots, K, \text{ if } \gamma_i = 1, \\ 0, & \text{if } \gamma_i = 0, \end{cases}$$

$$\gamma_i \sim \text{Bern}(\hat{\lambda}_i).$$

Note that we sample from discrete values of $\hat{\mu}_k$'s rather than the Gaussian distributions $\mathcal{N}(\hat{\mu}_k, \hat{\sigma}_k^2)$, as it incurs more memory cost to sample from $\mathcal{N}(\hat{\mu}_k, \hat{\sigma}_k^2)$, which is against the original purpose of DNNs compression. Another minor concern is that pruning via (posterior) distribution, although popular (Bai et al., 2020; Sun et al., 2022), fails to attain the exact target sparsity level due to its stochastic nature. As a consequence, it may require extra effort of second-round pruning. To handle this, one can adopt a semi-stochastic sampling scheme: instead of sampling $\gamma_i$ from the Bernoulli distribution with parameter $\hat{\lambda}_i$ independently, one can directly set $\gamma_i = 0$ for those who have the smallest $\hat{\lambda}_i$ values (i.e., smaller $\hat{\lambda}_i$ implies a higher chance of $\tilde{\theta}_i = 0$), and set the rest to be 1. In such a way, the model sparsity level is fully tunable The proposed inference procedure is summarized in algorithm 2.

---

**Algorithm 2** Inference Phase

---

**Input:** A sparse quantized weight distribution $\hat{q}(\tilde{\theta})$, Bayesian Model Average number $N$, and sparsity level $s_t$.
1: **for** $n \leftarrow 1$ to $N$ **do**
2:     Sample $\tilde{\theta}_q$ from the posterior, i.e., $\tilde{\theta}_{q,i} = \hat{\mu}_k$ w.p. $\phi_k(\hat{\theta}_i; \hat{\varpi}, \tau_1)$.
3:     Prune the top-$s_t * 100\%$ of weights, according to the $\hat{\lambda}_i$, to zero, and get one final sample $\tilde{\theta}^n$.
4: **end for**
5: Inferences via Bayesian Averaged Model, e.g., Bayesian prediction as $\hat{y} = \frac{f(x; \tilde{\theta}^n)}{N}$.
**Output:** Bayesian inferences such as $\hat{y}$.

---

**Additional Remark** While our approach described above uses one quantization set $\mathcal{Q}_A$ for all weight parameters $\theta_i$, extending our method to use layer-wise quantization sets is natural. That is, the group of weight parameters within one layer uses its own quantization set, and different layers have different trainable quantization sets. Our implementation in the next section always uses layer-wise quantization sets.

## 5 EXPERIMENTS

To demonstrate the effectiveness of our method, we consider various experiments and models. We test our methods on variants of the following models and tasks: ResNet (He et al., 2016) for image classification task on CIFAR-10/100 (Krizhevsky et al., 2009) and BERT (Devlin, 2018) for question answering task on SQuAD V1.1 (Rajpurkar, 2016). The Appendix A contains additional experiments and full details of our experiment settings. Our primary performance metrics for comparison purposes are the compression rate (CR) and the accuracy drop (Acc. Drop). Given a baseline model, i.e., full-precision pre-trained model, the former is the ratio between the baseline model's memory footprint and the compressed model's, and the latter measures the decline in predictive performance after compression. Note that the baseline model is also used as the initialization of $\theta$ in our algorithm.

### 5.1 CIFAR

In this section, we present experiments using ResNet architectures on the CIFAR-10 and CIFAR-100 datasets. When compressing ResNet models, our method requires fine-tuning over the training dataset, completing the compression process within 10 epochs. To achieve high compression rates, we represent each layer's weights with either 4 or 16 components (i.e. $K = 4$ or $K = 16$ for each layer) and apply a sparsity level of $50\%$. As shown in Table 1, our methods compress the models by factors ranging from $16 \sim 32\times$ while keeping accuracy drops below $1.3\%$. For example, compressing ResNet-20 by a factor of 16 results in an accuracy drop of only $0.52\%$. Likewise, compressing ResNet-32 by a factor of $32\times$ yields a minimal accuracy reduction of $1.29\%$. Additionally, we compress ResNet-56 by a factor of 32, observing an accuracy drop of only $0.84\%$. Compared to other methods, our approach achieves much higher compression rates with smaller decreases in accuracy.

Subsequently, we compress ResNet-18 and ResNet-50 models and evaluate them on the CIFAR-100 dataset, comparing our results with the DGMS (Dong et al., 2022) compression method. To investigate the effectiveness of our method in handling quantization noise and to ensure a fair comparison,

| Model | Method | Pruning/Quantization | Bits | NZ% | CR | Top-1 Acc. |
|---|---|---|---|---|---|---|
| ResNet-20 | FP32 Dense | NA | 32 | 100% | 1× | 92.60% |
| | Method | Pruning/Quantization | Bits↓ | NZ%↓ | CR↑ | Top-1 Acc. Drop ↓ |
| | LQNets | Q | 2 | 100% | 16× | 1.2% |
| | DGMS | P+Q | 2 | 55.6% | 28.8× | 0.87% |
| | **SQS(Ours)** | P+Q | 4 | 50% | 16× | 0.52% |
| | **SQS(Ours)** | P+Q | 2 | 50% | 32× | 1.47% |
| ResNet-32 | Method | Pruning/Quantization | Bits | NZ% | CR | Top-1 Acc. |
| | FP32 Dense | NA | 32 | 100% | 1× | 93.53% |
| | Method | Pruning/Quantization | Bits↓ | NZ%↓ | CR↑ | Top-1 Acc. Drop ↓ |
| | TTQ | Q | 2 | 100% | 16× | 1.9% |
| | DGMS | P+Q | 2 | 58.7% | 27.2× | 1.3% |
| | **SQS(Ours)** | P+Q | 2 | 50% | **32×** | **1.29%** |
| ResNet-56 | Method | Pruning/Quantization | Bits | NZ% | CR | Top-1 Acc. |
| | FP32 Dense | NA | 32 | 100% | 1× | 94.37% |
| | Method | Pruning/Quantization | Bits↓ | NZ%↓ | CR↑ | Top-1 Acc. Drop ↓ |
| | TTQ | Q | 2 | 100% | 16× | 1.06% |
| | L1 | P | 32 | 10% | 10× | 1.83% |
| | DGMS | P+Q | 2 | 51.8% | 30.9× | 0.89% |
| | **SQS(Ours)** | P+Q | 2 | 50% | **32×** | **0.84%** |

Table 1: Comparison across different compression methods for compressing ResNet Models on CIFAR-10. P+Q: joint pruning and quantization, P: pruning only, Q: quantization only, Bits: weights quantization bit-width, NZ%: proportion of non-zero parameter, CR: compression rate. FP32 Dense denotes the baseline full-precision model. Compared methods are LQNets (Zhang et al., 2018), TTQ (Zhu et al., 2017), L1 (Li et al., 2017) and DGMS (Dong et al., 2022).

we fixed the sparsity level at zero (i.e., the compression effect is fully due to weight sharing) and varied the number of Gaussian components. The fewer the components, the higher the compression and quantization error, and we assess the trade-off between compression and performance. As depicted in Figure 2, even when using only 8 Gaussian components, our method only incurs an accuracy drop of less than 1%. Moreover, our approach exhibits more robustness against the intrinsic noise introduced by the quantization than DGMS. As the number of Gaussian components decreased, leading to increased quantization noise, our method consistently outperformed DGMS.

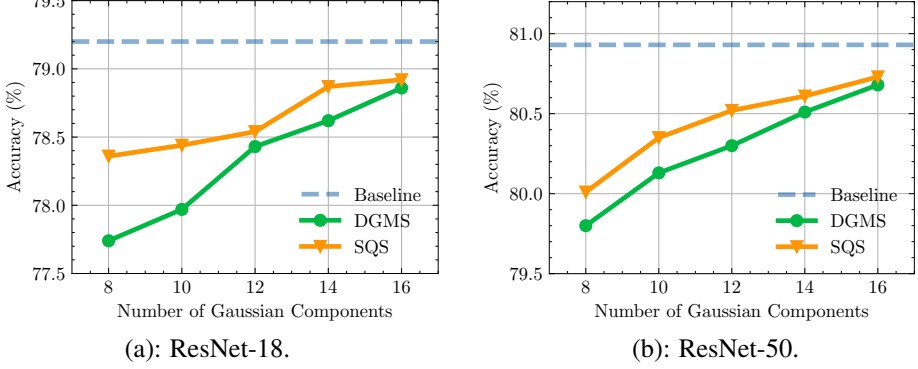

(a): ResNet-18.  (b): ResNet-50.

Figure 2: Accuracy of compressed ResNet-18 and ResNet-50 with CIFAR-100 dataset. (a): ResNet-18 Model. (b): ResNet-50 Model. With a large number of Gaussian components, our method is comparable to DGMS; however, with fewer Gaussian components, it achieves less performance degradation.

## 5.2 SQUAD

We further investigate our compression method on attention-based models. We apply our compression model on BERT (Devlin, 2018) base model and test it on the SQuAD V1.1 dataset (Rajpurkar, 2016). Similarly, we consider the F1 score drop and compression rate as the evaluation metrics. During the compression process, the BERT model is fine-tuned on the training dataset, with the entire procedure completed within 3 epochs.

We compressed the BERT model using $K = 16$ Gaussian components and pruned 75% of its parameters, leading to a $32\times$ compression rate. We employed layer-wise quantization combined with unstructured pruning to attain these results. Notably, our method resulted in an F1 score drop of only 1.66, which is less than that observed with existing methods, proving its superior performance retention despite the high compression rate.

| Model | Pruning/Quantization | Bits | NZ% | CR | F1 |
|---|---|---|---|---|---|
| FP32 Dense | NA | 32FP | 100% | $1\times$ | 88.68 |
| Method | Pruning/Quantization | Bits $\downarrow$ | NZ% $\downarrow$ | CR$\uparrow$ | F1 Drop $\downarrow$ |
| GMP | P | 32 | 50% | $2\times$ | 22.89 |
| L-OBS | P | 32 | 50% | $2\times$ | 10.86 |
| ExactOBS | P | 32 | 25% | $4\times$ | 6.43 |
| PLATON | P | 32 | 20% | $5\times$ | 2.2 |
| OBQ | Q | 3 | 100% | $10.7\times$ | 3.24 |
| GPTQ | Q | 3 | 100% | $10.7\times$ | 2.51 |
| OBC (ExactOBS+OBQ) | P+Q | 4 | 50% | $16\times$ | 2.33 |
| **SQS (Ours)** | **P+Q** | **4** | **25%** | **32$\times$** | **1.66** |

Table 2: Comparison across different compression methods for compressing BERT base model on the SQuAD V1.1. P+Q: joint pruning and quantization, P: pruning only, Q: quantization only, Bits: weights quantization bit-width, NZ%: proportion of non-zero parameter, CR: compression rate. FP32 Dense denotes the baseline full-precision model. Compared Methods are GMP (Zhu & Gupta, 2017), L-OBS (Dong et al., 2017), PLATON (Zhang et al., 2022a), GPTQ (Frantar et al., 2022), ExactOBS, OBQ and OBC (Frantar & Alistarh, 2022).

## 5.3 ABLATION STUDY

In this section, we conduct an ablation study to evaluate the impact of our proposed method. Specifically, we perform a detailed analysis of the effect of the spike-and-slab distribution. For comparison, we consider a zero-mean Gaussian distribution as the prior and replace the delta distribution with a Gaussian distribution in the variational family. That is, any $q'(\theta) \in \mathcal{F}'$ has the form:

$$\tilde{\theta}_i | \gamma_i \sim \gamma_i g\left((\mu_1, \sigma_1^2), \cdots, (\mu_K, \sigma_K^2), \theta_i\right) + (1 - \gamma_i)\mathcal{N}(0, \sigma_0^2),$$

$$\gamma_i \sim \text{Bern}(\tilde{\lambda}_i), \text{ for } i = 1, \ldots, T.$$

Based on this, we can get the modified marginal variational distribution $q'(\tilde{\theta}_i)$ as:

$$\tilde{\lambda}_i g\left((\mu_1, \sigma_1^2), \cdots, (\mu_K, \sigma_K^2), \theta_i\right) + (1 - \tilde{\lambda}_i)\mathcal{N}(0, \sigma_0^2). \tag{13}$$

Thus following the same reasoning and derivation the as we get the equation (10), we can have:

$$\Omega' = -\mathbb{E}_{q'(\tilde{\theta})}\left[\log p(\mathcal{D}; \tilde{\theta})\right] + \sum_{i=1}^{T} D_{\text{KL}}\left(q'(\tilde{\theta}_i)||\mathcal{N}(0, \sigma_0^2)\right)$$

$$= -\mathbb{E}_{q'(\tilde{\theta})}\left[\log p(\mathcal{D}; \tilde{\theta})\right] + \sum_{i=1}^{T} D_{\text{KL}}\left(q'(\tilde{\theta}_i)||(\tilde{\lambda}_i + (1 - \tilde{\lambda}_i))\mathcal{N}(0, \sigma_0^2)\right)$$

$$\leq -\mathbb{E}_{q'(\tilde{\theta})}\left[\log p(\mathcal{D}; \tilde{\theta})\right] + \sum_{i} \tilde{\lambda}_i D_{\text{KL}}\left(g((\mu_1, \sigma_1^2), \cdots, (\mu_K, \sigma_K^2), \theta_i)||\mathcal{N}(0, \sigma_0^2)\right). \tag{14}$$

We compressed a ResNet-18 model at varying sparsity levels, representing each layer's weights with 16 components, and evaluated it on the CIFAR-100 dataset, comparing the results with our proposed spike-and-slab prior method. As shown in Table 3, using a Gaussian prior to induce posterior sparsity on DNN weights does achieve reasonable performance at low sparsity levels. This is because, to minimize the term $-\mathbb{E}_{q'(\tilde{\theta})}\left[\log p(\mathcal{D}; \tilde{\theta})\right]$ in (14), important weights with larger magnitudes are assigned higher values of $\tilde{\lambda}_i$ which can guide effective pruning. However, this approach becomes insufficient when the sparsity is high, as the objective (14) does not favor high sparsity. In contrast, with the spike-and-slab distribution, the objective (10) includes an additional term $\sum_{i=1}^{T}\left(\tilde{\lambda}_i \log \frac{\tilde{\lambda}_i}{\lambda} + (1 - \tilde{\lambda}_i) \log \frac{1-\tilde{\lambda}_i}{1-\lambda}\right)$ which pushes the $(1 - \tilde{\lambda}_i)$ towards the desired sparsity level $(1 - \lambda)$, allowing the algorithm to better explore highly sparse weights. The results in Table 3 confirm that the spike-and-slab prior outperforms the Gaussian prior, particularly at higher sparsity levels.

| Prior | Bits | NZ% | CR | Top-1 Acc. |
|---|---|---|---|---|
| FP32 Dense | 32 | 100% | 1× | 79.26% |
| Prior | Bits↓ | NZ%↓ | CR↑ | Top-1 Acc. Drop↓ |
| | 4 | 50% | 16× | 4.51% |
| Gaussian | 4 | 40% | 20× | 5.6% |
| | 4 | 30% | 26.6× | 11.42% |
| | 4 | 20% | 40× | 44.04% |
| | 4 | 50% | 16× | 3.12% |
| Spike-and-slab | 4 | 40% | 20× | 3.21% |
| | 4 | 30% | 26.6× | 5.54% |
| | 4 | 20% | 40× | 5.59% |

Table 3: Comparison of Gaussian prior and Spike-and-slab prior for compressing a ResNet-18 model on the CIFAR-100 dataset. Bits: weights quantization bit-width, NZ%: proportion of non-zero parameter, CR: compression rate. FP32 Dense denotes the baseline full-precision model. Using Gaussian prior could provide reasonable performance when NZ% is low but fails when NZ% is less than 30%.

## 6 CONCLUSION

In this paper, we proposed a unified framework for compressing deep neural networks (DNNs) by combining pruning and quantization into one integrated optimization process through variational inferences. Our approach addresses the limitations of sequential pruning and quantization methods by exploring a broader solution space, enabling more efficient compression with minimal performance degradation. Additionally, by leveraging Bayesian model averaging which is robust to the quantization noise, we enhance the model's resilience to potential performance degradation. We demonstrated the effectiveness of our method on multiple datasets, including CIFAR-10/100 and SQuAD which supports that our method not only improves performance but also provides a more robust solution for compressing modern DNNs. Our results outperform existing methods in both compression rates and accuracy retention, making it a promising direction for efficient model compression in resource-constrained environments.

In future work, we aim to conduct theoretical analysis to bridge the gap between theory guarantees and empirical successes. We also plan to test our method on computationally demanding models, such as large-scale language models.

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

## A   MORE EXPERIMENT RESULTS

In this section, we provide additional details about our empirical experiments and results. During our experiments on CIFAR-10 and CIFAR-100, we compressed the ResNet variants within 10 epochs. Besides, we employed different learning rates for the parameter $\tilde{s}$ and the other parameters (0.015 for $\tilde{s}$ and $5 \times 10^{-5}$ for the others). The hyperparameters were set to $\tau_1 = 0.001$, $\tau_2 = 0.012$ and $\lambda = 0.01$. We selected $N = 10$ for model averaging. The runtime details are reported in Table 4. Additionally, we present our compression results with ResNet-18 on CIFAR-100, where each layer is represented by $T = 16$ components and $80\%$ of the parameters have been pruned.

|      | ResNet-18 | ResNet-20 | ResNet-32 | ResNet-50 | ResNet-56 |
|------|-----------|-----------|-----------|-----------|-----------|
| Time | 35.74min  | 32.48min  | 32.5min   | 35.75min  | 32.5min   |

Table 4: Runetime in minutes of Compression procedure on ResNet architecture tested on NVIDIA V100.

During the compression of the BERT model, we also employed different learning rates for the parameter $\tilde{s}$ and the other parameters, using $0.01$ for $\tilde{s}$ and $2 \times 10^{-5}$ for the rest. The hyperparameters were configured as $\tau_1 = 0.005$ and $\tau_2 = 0.01$. The compression procedure is finished within 3 epochs.

| Model | Method | Bits | NZ% | Top-1 Acc. | Top-1 Acc. Drop |
|-------|--------|------|-----|------------|-----------------|
| ResNet-18 | FP32 Dense | 32 | 100% | 79.26% | NA |
|           | Ours | 4 | 20% | 76.07% | 3.19% |

Table 5: Compresson Result of ResNet-18 on CIFAR-100. Bits: weights quantization bit-width, NZ%: proportion of non-zero parameter.

We also tested our method on the GPT-2 model (Radford et al., 2019), using perplexity as the evaluation metric on the Penn Treebank (Taylor et al., 2003) dataset. Perplexity measures how well a language model predicts a sequence of words; lower perplexity indicates better predictive performance and a higher level of certainty in the model's predictions. While compressing the GPT-2 model, we set the learning rates for $\tilde{s}$ to $0.01$ and $2 \times 10^{-5}$ for the rest. The hyperparameters were set to $\tau_1 = 0.00001$ and $\tau_2 = 0.01$. The performance result is reported in Table 6. We compressed GPT-2 by a factor of $38.53$ and achieved a satisfactory perplexity of $30.80$.

| Method | Compression Rate | Perplexity $\downarrow$ |
|--------|------------------|------------|
| Ours   | $38.53\times$    | 30.80      |

Table 6: Compression Result of GPT-2 on Penn Treebank.

