# OpenReview forum: "Efficient Bayesian DNN Compression through Sparse Quantized Sub-distributions"
_ICLR.cc/2025/Conference — Submitted to ICLR 2025_

### Official Review · Reviewer_kfPy · 2024-10-24

**Soundness:** 3
**Presentation:** 3
**Contribution:** 3
**Rating:** 6
**Confidence:** 4

**Summary:**

This paper proposes a framework that integrates quantization and pruning into a unified process, rather than applying them sequentially. This study reformulates the weight quantization issue as the selection of a sub-distribution within the Gaussian Mixture Model, while the pruning problem is approached as a probability sampling problem from a Bernoulli Distribution. The authors established an approximate ELBO objective that consolidates the optimization processes for pruning and quantization into a unified optimization framework. Results for the ResNet and BERT models in image classification and question answering tasks demonstrate a significant compression rate with low performance degradation.

**Strengths:**

1. The idea of combining the optimization processes of quantization and pruning into a unified optimization space is interesting, and the optimization objectives have been thoroughly theoretically derived.
2. The paper is easy to read.

**Weaknesses:**

1. While the authors perform experiments on ResNet and BERT models, they restrict their testing to small-sized datasets (CIFAR10/100, SQUAD V1.1). Furthermore, the evaluated tasks are standard (image classification), in contrast to the previous work DGMS compared in this paper, which conducted experiments on a large dataset (ImageNet) and an object detection task. I think the proposed method may have certain inherent limitations.
2. The idea is not novel.
3. The main limitation of this paper is that proposed method lacks more experimental results on larger datasets (i.e., Imangenet, COCO2017, VOC), larger networks (i.e., VIT, EfficientNet, LLMs, etc), and compared with more competitors (i.e., OMPQ [1], EMQ [2], etc).
4. The statement "HAQ is suboptimal" is not accurate in Section I. You should provide empirical results to prove your statement.
5. The format of references is not correct, i.e.,  "Deep learning with limited numerical precision"
6. There exist some typos/grammatical errors in the paper and should be revised.
7. The presentation of the paper is bad.
8. The paper claims that the method outperforms state-of-the-art methods, but a more comprehensive comparative analysis is needed. Detailed comparisons with other existing approaches, along with discussions about the reasons behind the performance differences, would strengthen the argument for the superiority.
9. Exploring the reasons behind the success of these techniques and providing intuitive explanations would contribute to the overall scientific contribution of the work.
10. Provide a more thorough discussion of the generalization ability, robustness, and potential applications of the proposed approach.



[1] OMPQ: Orthogonal Mixed Precision Quantization
[2] EMQ: Evolving Training-free Proxies for Automated Mixed Precision Quantization

**Questions:**

1. In line 176, is $T$ used before definition?

---

> ### Author Response · Authors · 2024-11-28
> **Rebuttal by Authors**
>
> Thank you for your valuable feedback. We collect the response to address your concerns as follows.
>
> ## 1. Small Dataset, Small Model
>
> Thank you for pointing this out. We acknowledge that our experiments are currently limited to smaller models and datasets due to constraints in computational resources. However, our proposed method is designed to be scalable, and we believe it can be applied to larger models and datasets as well. We plan to conduct experiments on larger-scale models and datasets in future work to further validate the scalability and effectiveness of our approach.
>
> ## 2. The idea is not novel.
>
> Thank you for your comment. While we acknowledge that the idea of combining pruning and quantization is not novel, our approach introduces a significant advancement. Specifically, we unify pruning and quantization through variational inference, which allows us to explore a global optimization space rather than applying these techniques sequentially in a greedy manner, as done in previous works. This unified optimization framework is novel and enables more effective compression of deep neural networks.
>     In summary, we propose a novel Bayesian variational learning approach that integrates pruning and quantization into a single framework, achieving a unified solution for efficient DNN compression.
>     By leveraging spike-and-slab priors and Gaussian Mixture Models, we identify optimal sparse quantized sub-distributions, enhancing both compression efficiency and robustness to quantization-induced noise through Bayesian model averaging.
>     We demonstrate the effectiveness of our method on multiple datasets, achieving state-of-the-art compression rates with minimal performance degradation.
>
> ## 3. More experimental results & Compared with more competitors
>
> Thank you for pointing out the relevant references and methods for comparison. We agree that including comparisons with state-of-the-art techniques would significantly enhance the evaluation of our method. Due to time and computational limitations, we were unable to conduct these additional comparisons in this version. However, we are committed to incorporating them in future work to provide a more thorough assessment of our approach against recent advancements. We appreciate your suggestion, as it will help improve the overall quality and impact of our research.
>
> ## 4. The statement "HAQ is suboptimal" is not accurate
>
> Thank you for pointing out the inaccurate wording. We have replaced it as:
> "[1] proposed a model compression pipeline that sequentially applies pruning and weight quantization, achieving significant compression rates without sacrificing much accuracy, however, the sequential application fails to explore the complementarity of pruning and quantization~[2]."
>
> [1]: Song Han, Huizi Mao, and William J Dally. Deep compression: Compressing deep neural networks
> with pruning, trained quantization and huffman coding.
>
> [2]: Shipeng Bai, Jun Chen, Xintian Shen, Yixuan Qian, and Yong Liu. Unified data-free compression:
> Pruning and quantization without fine-tuning.
>
> ## 5. Format of references
>
> Thank you for pointing this out. We appreciate your attention to detail and will carefully review and correct the format of all references, including "Deep learning with limited numerical precision," to ensure they adhere to the required standards in the revised version of the paper.
>
> ## 6. Typos/grammatical errors
>
> Thank you for pointing this out. We appreciate your careful review and will thoroughly proofread the paper to address any typos and grammatical errors in the revised version. Ensuring clarity and correctness is a priority for us.
>
> ## 7. The presentation of the paper is bad.
>
> Thank you for your feedback. We will thoroughly proofread and polish the paper to improve its presentation.
>
> ## 8. Further Validation of Superior Performance
>
> Thank you for the valuable feedback. We agree that a more comprehensive comparative analysis would further strengthen our argument. Our method introduces a unified optimization process that, unlike sequential or greedy compression approaches, provides greater flexibility in exploring the complementarity between pruning and quantization. Additionally, we propose a Bayesian model averaging approach to enhance robustness against quantization noise. The rationale is that averaging can mitigate random noise, making the model more resilient to the effects of quantization.
> We recognize the importance of providing detailed comparisons and in-depth discussions about the performance differences with other approaches.  Thank you again for pointing this out.

---

> ### Author Response · Authors · 2024-11-28
> **Rebuttal by Authors (Cont.)**
>
> ## 9. Exploring the reasons behind the success of these techniques and providing intuitive explanations would contribute to the overall scientific contribution of the work.
>
> Thank you for your insightful suggestion! To address this, we have included an ablation study in Section 5.3 to analyze the impact of the spike-and-slab prior. Our results demonstrate that, compared to a Gaussian prior, the spike-and-slab distribution is more effective in achieving high sparsity. This advantage stems from the term $\sum_{i=1}^T \left(\tilde{\lambda}_i \log\frac{\tilde{\lambda}_i}{\lambda} + (1-\tilde{\lambda}_i) \log\frac{1-\tilde{\lambda}_i}{1-\lambda} \right)$ in the objective, which encourages the weights to align with the desired sparsity level. We hope this addresses your concern and provides clarity on the impact of different prior choices. We have also shown in Figure 2, that Bayesian averaging is robust to quantization thus further enhancing our methods.
>
> ## 10. Provide a more thorough discussion of the generalization ability, robustness, and potential applications of the proposed approach.
>
> Thank you for highlighting this important aspect. We agree that discussing the generalization ability, robustness, and potential applications of our proposed approach would enhance the paper.
> Generalization Ability: Our method is designed to generalize across different architectures and tasks due to its unified framework for pruning and quantization. By leveraging variational inference and spike-and-slab priors, the approach is adaptable to diverse model types, from convolutional networks to transformer-based architectures, as demonstrated in our experiments.
> Robustness: The Bayesian model averaging component of our method plays a critical role in enhancing robustness. By mitigating the impact of quantization-induced noise, the approach maintains performance across varying sparsity and bit-width levels. This property makes the method reliable even in extreme compression scenarios.
> Potential Applications: Our framework has broad applicability in scenarios requiring efficient deployment of deep learning models. For instance, it is well-suited for edge devices with limited computational and storage resources.
> We will expand on these aspects in the revised version to provide a more thorough discussion and better highlight the practical and scientific contributions of our approach. Thank you again for your valuable feedback.
>
> ## 11. In line 176, is $T$ used before definition?
>
> Thank you for pointing out. $T$ is defined as the total number of weights in the later context of the paper.
> We have changed it to the following for a better understanding.
> "Let $\theta$ denote the weights vector in $R^T$ indicating the set of weights, with $T$ being the total number of weights in the DNN.
> Rather than storing $T$ full-precision weights, the weights are quantized into a few discrete values (i.e., shared full precision value), and only a small index indicating which shared value in $\mathcal{Q}_A$ is assigned is stored for each weight, where the $T$ is the total number of weights."

---

> > ### Comment · Reviewer_kfPy · 2024-11-29
> >
> > Thanks for your reply. I will increase my score. However, the problems, i.e., larger datasets are needed, exploring the reasons behind the success of these techniques and providing intuitive explanations (visualization). All comments aim to enhance the reader's understanding and clarity of the paper. I advise the authors to address all remarks.

---

### Official Review · Reviewer_ECZE · 2024-11-03

**Soundness:** 3
**Presentation:** 3
**Contribution:** 2
**Rating:** 6
**Confidence:** 3

**Summary:**

This paper introduces a novel method that simultaneously achieves model pruning and low-bit quantization through Bayesian variational inference, effectively compressing deep neural networks with minimal performance degradation. This method explores a unified optimization space using a spike-and-slab prior combined with Gaussian Mixture Models, achieving significant (up to 32x) compression rates with negligible accuracy loss on CIFAR-10, CIFAR-100, and SQuAD datasets. The proposed approach outperforms existing techniques and demonstrates that Bayesian model averaging can further mitigate quantization noise for more robust compressed models.

**Strengths:**

Pros:
1.	The proposed method unified the pruning and quantization, further shrinking the model size with less model accuracy drop.
2.	Approximating the sub-distribution problem and leveraging the variational learning to solve the problem reduces the complexity of optimizing the quantization and sparsity in all weight data.
3.	Bayesian average training is introduced to enhance the performance.

**Weaknesses:**

Cons:
1.	The experiments are not sufficient to demonstrate the effectiveness of the LLM model. The Bert model and GPT-2 model are not widely used LLM models.
2.	The baselines are not the SOTA work. Some SOTA works, e.g. AWQ, are expected to be compared in evaluation.
3.	Can you provide the evaluation in Table 5 with some baselines to show the proposed framework outperforms the existing works?

**Questions:**

See weeknesses

---

> ### Author Response · Authors · 2024-11-28
> **Rebuttal by Authors**
>
> Thank you for your valuable review and feedback, which have provided us with an opportunity to clarify and strengthen our work.
>
> ## 1. The experiments are not sufficient to demonstrate the effectiveness of the LLM model. The Bert model and GPT-2 model are not widely used LLM models.
>
> Thank you for your feedback. We agree that including more widely-used LLMs could further strengthen our results. We selected BERT and GPT-2 as representative models due to their accessibility and the computational resources available to us. However, our method maintains the same computational complexity in terms of the number of operations, indicating that it should scale efficiently to larger models, and we believe it can generalize well to larger and more recent LLMs.
>
> ## 2. The baselines are not the SOTA work.
>
> Thank you for highlighting this point. We acknowledge that including comparisons to state-of-the-art methods like AWQ would further strengthen our evaluation. Due to computational and time constraints, we were unable to run comparisons with all SOTA methods. However, we will certainly aim to include such comparisons in future work to provide a more comprehensive evaluation of our approach against the latest advancements. Your suggestion is greatly appreciated, and we agree that it will enhance the impact of our findings.
>
> ## 3. Can you provide the evaluation in Table 5 with some baselines to show the proposed framework outperforms the existing works?
>
> Thank you for your insightful suggestion. We agree that adding more baselines in Table 5 would provide a clearer demonstration of how our proposed framework compares to existing works. Due to time constraints, we were unable to include all relevant baselines in the current submission. However, we plan to address this in future revisions to better highlight our method's performance relative to other approaches.

---

### Official Review · Reviewer_9LAB · 2024-11-04

**Soundness:** 4
**Presentation:** 3
**Contribution:** 3
**Rating:** 6
**Confidence:** 4

**Summary:**

This paper presents a novel method called Sparse Quantized Sub-distribution (SQS) that unifies neural networks pruning and quantization through Bayesian variational inference. The key innovation is treating both compression techniques as part of a single optimization problem rather than sequential steps. The method leverages spike-and-slab priors combined with Gaussian Mixture Models (GMMs) to achieve both network sparsity and low-bit representation, while using Bayesian model averaging to improve robustness to quantization noise.

**Strengths:**

- The unified treatment of pruning and quantization as a single single optimization problem is novel and well-motivated.
- The mathematical framework combining spike-and-slab priors with GMMs is theoretically sound.
- The preliminaries are rigorously explained.
- Mathematical notation is consistent and well explained.
- The methodology is presented in a reproducible manner.
- Strong results for ResNet/BERT on CIFAR/SQuAD. Upto 32x compression rates with minimal performance degradation. The baselines are well defined and diverse for a fair comparison.

**Weaknesses:**

- Some hyperparameter choices could be better justified.
- The results presented are technically on just 2 different architectures : ResNet and BeRT, which are very limited.

Based on the suggestions/questions below, I am more than willing to increase my score as the paper does show promise.

**Questions:**

1. I would like to see similar results across other computer vision models, on other datasets than just CIFAR. Maybe include results on ImageNet1k with a wider variety of models.
2. Similarly for the NLP task, I would like to see performance on other tasks like GLUE/SuperGLUE/MMLU etc with a wider variety of models and not just BERT.
3. Consider adding an ablation study on the impact of different prior choices other than spike-and-slab. It might not work and that is fine, but having a better understanding on why only this prior works would be helpful.
4. An analysis on scalability to larger models would also be helpful.
5. For Table 1 and Table 2, consider adding in the captions the type of test/dataset on which you evaluate the model on. Otherwise the captions and the results are a bit ambiguous if one does not go over the experiment section in detail.

---

> ### Author Response · Authors · 2024-11-28
> **Rebuttal by Authors**
>
> We deeply appreciate your detailed review. Your observations are invaluable, and we address your concerns as follows:
>
> ## 1 Some hyperparameter choices could be better justified
> Thank you for your feedback! We have provided detailed hyperparameter settings in Appendix A and included explanations for their selection. We hope this addresses your concern, and we are happy to elaborate further if needed.
>
> ## 2 Similar results across other computer vision models
> Thank you for your valuable feedback! Due to limitations in computational resources and time, we were unable to include a broader range of models and larger datasets in this study. We appreciate your suggestion and plan to extend our experiments in future work to include a wider variety of models, such as ViT, and evaluate on larger datasets like ImageNet1k to provide a more comprehensive analysis.
>
> ## 3 Other tasks like GLUE/SuperGLUE/MMLU etc with a wider variety of models and not just BERT
> Thank you for the suggestion! Similar to vision tasks,  due to limitations in computational resources and time, we were unable to include a broader range of models and larger datasets in this study. We also plan to explore other tasks such as GLUE, SuperGLUE, and MMLU in future work, using a wider variety of models beyond BERT (larger models) to provide a more comprehensive evaluation.
>
> ## 4 Ablation study
> Thank you for your suggestion! We have included an ablation study to examine the effect of the spike-and-slab prior, which can be found in Section 5.3. In summary, our results indicate that compared to a Gaussian prior, the spike-and-slab distribution is better suited for achieving high sparsity. This advantage arises from the term $\sum_{i=1}^T \left(\tilde{\lambda}_i \log\frac{\tilde{\lambda}_i}{\lambda} + (1-\tilde{\lambda}_i) \log\frac{1-\tilde{\lambda}_i}{1-\lambda} \right)$ in the objective, which encourages the weights to align with the desired sparsity level. We hope this addresses your concern and provides clarity on the impact of different prior choices.
>
> ## 5 Analysis of scalability to larger models
> Thank you for your feedback regarding scalability analysis. Our method does not increase the computational complexity in terms of the order of operations required, which suggests that it is scalable to larger models. The reason we did not include larger models in our experiments is due to limited computational resources. However, we are confident that our approach can be extended to larger architectures, and we plan to explore this in future work.
>
> ## 6 Table Caption:
> Thank you for pointing this out! We have updated the captions for Table 1 and Table 2 to include the type of test/dataset used for evaluation, following your suggestion.

---

> > ### Comment · Reviewer_9LAB · 2024-12-01
> > **Official Comment by Reviewer 9LAB**
> >
> > I thank the authors for the rebuttal. All my concerns have been resolved and I would like to keep my score.

---

### Official Review · Reviewer_Vowk · 2024-11-04

**Soundness:** 2
**Presentation:** 2
**Contribution:** 2
**Rating:** 5
**Confidence:** 3

**Summary:**

The authors proposed a variational inference-based method for pruning and quantizing DNNs.

**Strengths:**

+ Well-motivated idea.
+ Novel ELBO approximation enabling the method in practice.
+ Experiments on both vision and natural language networks.

**Weaknesses:**

- Computational expenses and data requirement not well discussed.
- The existence of extra hyperparameters requires further tuning, and the nontrivial balance between the aggressiveness of pruning and of quantization is not well addressed.

**Questions:**

* It is not really clear to a practitioner, after reading the paper, how to tune the knobs of this new method, at what cost, and how to factor in the specific information of the target hardware the model is to be deployed on.

**Details Of Ethics Concerns:**

None.

---

> ### Author Response · Authors · 2024-11-28
> **Rebuttal by Authors**
>
> Thank you for your valuable feedback. We collect the response to address your concerns as follows.
>
> ## 1 Computational expenses and data requirement not well discussed.
>
> We appreciate your valuable feedback regarding the discussion of data and computation requirements. Indeed, our method requires retraining the model on the full training dataset. However, the compression process can be finished within relatively small epochs. We added the following discussion in the revised PDF.
>     "When compressing ResNet models, our method requires fine-tuning over the training dataset, completing the compression process within $10$ epochs."
>     "During the compression process, the BERT model is fine-tuned on the training dataset, with the entire procedure completed within $3$ epochs."
>
> ## 2 Existence of extra hyperparameters requires further tuning, tradeoff between Pruning and Quantization.
>
> Thank you for your thoughtful feedback regarding the effort required for hyperparameter tuning. We understand that this can be a concern, and we truly appreciate the opportunity to clarify. Hyperparameter tuning is indeed a fundamental part of modern machine learning workflows, and we have designed our method to align with this standard practice. By integrating pruning and quantization into a unified optimization process, our approach helps reduce the manual effort typically needed to balance these two operations, as shown in our experiments. We hope this addresses your concern, and we look forward to continuing to refine our approach to make it even more user-friendly in the future.
>
> ## 3 How to tune
> Thank you for highlighting this important point about the practical usability of our method. We recognize the importance of providing clear guidance for practitioners on tuning hyperparameters and considering hardware-specific constraints.
> Regarding the inclusion of hardware-specific factors, we agree that this is a critical consideration for real-world deployment. While our current work focuses on general compression methods, extending it to account for target hardware characteristics is an exciting direction for future research.
>  We plan to conduct additional experiments to analyze the interplay between pruning and quantization in the future.

---

### Public Comment · ~Thomas_Pfeil1 · 2025-10-13

How does this work relate to
Achterhold, Jan, et al. "Variational network quantization." ICLR. 2018.
https://openreview.net/forum?id=ry-TW-WAb
?

---

### Meta-Review · Area_Chair_FpfZ · 2024-12-21

**Metareview:**

**Summary**

This paper introduces Sparse Quantized Sub-distribution (SQS), a novel framework that integrates model pruning and low-bit quantization into a unified optimization process using Bayesian variational inference. By treating weight quantization as selecting sub-distributions within a Gaussian Mixture Model and pruning as sampling from a Bernoulli Distribution, the method leverages spike-and-slab priors combined with GMMs. This approach consolidates the optimization processes for pruning and quantization, allowing for significant network compression (up to 32x) with minimal performance loss, as demonstrated on ResNet and BERT models across CIFAR-10, CIFAR-100, and SQuAD datasets. The method outperforms existing techniques by mitigating quantization noise through Bayesian model averaging, enhancing the robustness of the compressed models.



**Strengths**


* The paper is well-written and easy to follow.
* The unified treatment of pruning and quantization as a single optimization problem is well-motivated, resulting in shrinking the model size with less model accuracy drop.
* Approximating the sub-distribution problem and leveraging the variational learning to solve the problem reduces the complexity of optimizing the quantization and sparsity in all weight data.
* The mathematical framework combining spike-and-slab priors with GMMs is theoretically sound and interesting.

**Weaknesses**

* While the authors perform experiments on ResNet and BERT models, they restrict their testing to small-sized datasets (CIFAR10/100, SQUAD V1.1).  The proposed method lacks more experimental results on larger datasets for the results to be conclusive.
* The baselines used in this paper are not the current state-of-the-art methods.
* The paper's presentation could be significantly improved.

**Conclusion**

The paper received borderline reviews, primarily due to the lack of robust baselines and substantial scale experiments, as noted by most reviewers. After reviewing the manuscript, the feedback from the reviewers, and the authors’ responses, I recognize the merit in the approach of integrating pruning and quantization through variational inference using spike-and-slab priors and Gaussian Mixture Models. However, the limited scope of the experiments and the inadequate comparison with more recent relevant literature make it challenging to assess the practical impact of this work. Based on these considerations, I recommend rejection as the paper does not seem ready for publication.

**Additional Comments On Reviewer Discussion:**

Despite my efforts to engage the reviewers in a discussion during the review period to reach a consensus on the paper’s merits and shortcomings, there was no participation in any discussion.

The paper was evaluated as borderline. While it exhibits methodological strengths, the conducted experiments were underwhelming. Given its potential for significant improvement, I recommend rejecting it in its current form.

---

### Decision · Program_Chairs · 2025-01-22

Reject